# Inflammation in Development and Aging: Insights from the Zebrafish Model

**DOI:** 10.3390/ijms25042145

**Published:** 2024-02-10

**Authors:** Marta Mastrogiovanni, Francisco Juan Martínez-Navarro, Teresa V. Bowman, María L. Cayuela

**Affiliations:** 1Department of Developmental and Molecular Biology, Albert Einstein College of Medicine, Bronx, NY 10461, USA; 2Gottesman Institute for Stem Cell Biology and Regenerative Medicine, Albert Einstein College of Medicine, Bronx, NY 10461, USA; 3Grupo de Telomerasa, Cáncer y Envejecimiento, Hospital Clínico Universitario Virgen de la Arrixaca, 30120 Murcia, Spain; 4Instituto Murciano de Investigación Biosanitaria-Arrixaca, 30120 Murcia, Spain; 5Centro de Investigación Biomédica en Red de Enfermedades Raras (CIBERER), ISCIII, 30100 Murcia, Spain

**Keywords:** *Danio rerio*, zebrafish, inflammation, aging, development, immunity, immune signaling, hematopoiesis, inflammaging

## Abstract

Zebrafish are an emergent animal model to study human diseases due to their significant genetic similarity to humans, swift development, and genetic manipulability. Their utility extends to the exploration of the involvement of inflammation in host defense, immune responses, and tissue regeneration. Additionally, the zebrafish model system facilitates prompt screening of chemical compounds that affect inflammation. This study explored the diverse roles of inflammatory pathways in zebrafish development and aging. Serving as a crucial model, zebrafish provides insights into the intricate interplay of inflammation in both developmental and aging contexts. The evidence presented suggests that the same inflammatory signaling pathways often play instructive or beneficial roles during embryogenesis and are associated with malignancies in adults.

## 1. Zebrafish as Inflammation Model

The zebrafish has become a prominent vertebrate animal model. Zebrafish are useful for studying human diseases as they share homology with more than 70% of human genes [1], and most pathways, cell types, and tissues involved in human diseases are conserved in zebrafish. Together with unquestionable advantages compared to other vertebrate models, such as rapid development, high fecundity, transparency, and ease of gene editing, the zebrafish is a powerful animal model to study a great variety of human diseases.

Some of the fields in which zebrafish have proven particularly valuable are the study of the role of inflammation in host defense, immune responses, development, tissue regeneration, and aging [2,3,4,5,6,7]. Inflammation can be triggered by several sources, including biological, chemical, and mechanical, and these have been largely employed in zebrafish models to address molecular and cellular questions, particularly in pathogenic conditions.

Because of the early emergence of innate immune cells [8] and the ease of assessing local and systemic microinjections, zebrafish have been established as a powerful model for numerous bacterial, viral, and fungal pathogens [9,10,11]. These infection models have advanced our understanding of host-pathogen interactions, cellular immunity, and emergency granulopoiesis, as well as the bases of pathogenesis and cell biology [11,12].

Zebrafish has also been demonstrated to be a reliable model system for studying human viral pathologies, including the most recent coronavirus disease 2019 [13]. For example, using an adult zebrafish model, it was observed that intranasally delivered SARS-CoV-2 S spike protein caused severe olfactory histopathology, providing key insights for the evaluation of the mechanisms of action and side effects of human intranasally delivered vaccines against SARS-CoV-2 [14]. Due to the unique characteristics of zebrafish, such as their small size, optical transparency, and simple external development, the zebrafish model system allows for the rapid screening of chemical compounds that affect inflammation. Likewise, Zebrafish has been employed to assess inflammation and toxicity upon chemical damage, including exposure to alcohol [15], nicotine [16], neuropeptides [17], and acids [18], enabling the identification of potential lead compounds for anti-inflammatory therapies.

Moreover, zebrafish fin injury models are robust tools for dissecting the role of inflammation in tissue damage and assessing regeneration and tissue repair. Indeed, the tailfin injury zebrafish model allows for evaluating the molecular mechanisms underlying the inflammatory response and drug discovery studies [19]. Furthermore, zebrafish fin injury models, in combination with zebrafish transgenic reporter lines of different cell types and/or chemical mediators of wound inflammation, also allow exploring the role of inflammation in the wound regenerative process and the impact of the immune cell migration machinery on the trajectory of inflammation [20,21,22,23]. Indeed, the alteration of leukocyte dynamics is considered an inflammation parameter [24,25], and tailfin injury in zebrafish represents a powerful physiological setup for qualitative and quantitative live-cell dynamics in inflamed tissues. 

Owing to the toolkits and putative applications listed above, zebrafish have been largely exploited for the assessment of inflammation cues and pathways in different setups. This review focuses on the use of zebrafish as a model to explore the role of inflammation in both development and aging.

## 2. Inflammatory Pathways Are Involved in Zebrafish Development and Aging

Inflammation implies the activation of a molecular and cellular response. The most significant mediators of the inflammatory response are cells of the innate immune system, such as macrophages and neutrophils. The immune system is highly conserved between zebrafish and humans [26,27,28]. Orthologs of the main proinflammatory cytokines involved in inflammatory responses, such as interleukin IL-6, tumor necrosis factor-α (TNF-α), and interleukin-1β (IL-1β), are all found in zebrafish [29]. The innate branch of the immune system matures first during zebrafish development [8]. Thereafter, the lymphoid lineage begins to mature in the thymus and in the kidney marrow, which is the equivalent of bone marrow in mammals [30,31]. The complete maturation of the adaptive immune system occurs at the juvenile stage [32], which means that zebrafish offer a unique and powerful model for assessing inflammation driven by innate immunity independently of adaptive responses at early stages. The following provides an overview of the molecular pathways and cytokines involved in sterile inflammation, as well as recent research findings that have shed light on the role of these signaling pathways in both development and aging.

### 2.1. NF-κB

The nuclear factor-κB (NF-κB) family is a group of inducible transcription factors that control both innate and adaptive immune responses, the expression of proinflammatory genes, the activation of inflammasomes [33], and the activation, differentiation, and effector function of inflammatory T cells [34]. The NF-κB family consists of five protein monomers (NF-κB1 p50, NF-κB2 p52, p65 [RelA], Rel B, and c-Rel), which can form up to 15 different homo- or heterodimers with distinct functions. Some of these exhibit transcriptional activity (NF-κB1 p50-p65 and NF-κB2 p52-RelB) [35]. These homo- or heterodimers normally exist as components of inactive cytoplasmic complexes bound by members of the inhibitor of κB (IκB) family. NF-κB activation occurs via two main signaling pathways: canonical and non-canonical [36]. Upon infection, the canonical NF-κB pathway or NF-κB essential modulator (NEMO, also known as IKKγ)-dependent pathway is activated. Canonical NF-κB family members include NF-κB1 p50, p65, and c-Rel. In most cells, p65 is the predominant NF-κB transcription factor and consequently drives the expression of many proinflammatory genes that control immunity and immune functions.

In addition to regulating immune and inflammatory responses, NF-κB is essential for embryogenesis, including dorsal–ventral patterning and limb, liver, skin, lung, neural, notochord, muscle, skeletal, and hematopoietic formation [37]. Defects in these components can lead to embryonic lethality [38,39] or developmental abnormalities [40,41]. Interestingly, although the NF-κB pathway is well conserved in mammals and fish [42], mice deficient in p65 die after 15–16 days of embryogenesis due to massive liver destruction [38], whereas p65-null zebrafish survive [43]. Therefore, p65-null zebrafish provide a crucial tool for addressing the specific function of NF-κB during embryogenesis, which has not been addressed in other animal systems.

NF-κB signaling has been shown to be important for other functions during the larval stages of zebrafish. For example, NF-κB-dependent activation in phagocytes occurs rapidly after myelin injury and is required for myelin debris degradation, inflammation resolution, and the initiation of the generation of new oligodendrocytes [44]. Moreover, blockage of NF-κB activity by overexpression of the dominant-negative form of the murine IκBα gene affects embryonic dorsalization and leads to notochord deformities [42].

In adults and aging, using zebrafish lines expressing the beta-cell-specific fluorescence ubiquitination cell cycle indicator (FUCCI), it has been shown that NF-kB signaling is activated in beta cells. These findings indicate that the activation of NF-kB is associated with a gradual decrease in beta cell proliferation as individuals age, accompanied by immune cell infiltration in the islets of Langerhans [6]. This study showed that zebrafish is a novel model for investigating the interconnections between aging, beta cell biology, the innate immune system, and diabetes.

Moreover, knockout of the longevity gene *sirt1* in zebrafish results in oxidative damage, chronic inflammation, and a shortened lifespan [45]. The expression of inflammation-related genes, such as *il1b*, *il6*, *nfkb*, *tnfa*, and *inos*, was increased in adult zebrafish with the *sirt1* mutation, showing that this upregulation occurs through NF-kB activation in the aorta during aging. Additionally, these mutant zebrafish exhibit inflammatory cell infiltration in neighboring organs, including the pancreas, intestine, and spleen.

In addition to NF-κB, other inflammatory pathways have been described in zebrafish developmental and aging studies. For example, complement-mediated inflammation has recently been shown to be involved in zebrafish myelogenesis [46]. The small GPI-anchored protein Cd59, a cell surface protein known to suppress complement-mediated inflammation, modulates Schwann cell (SC) proliferation during development [46].

### 2.2. TNF-α

TNF-α is a pleiotropic inflammatory cytokine that is produced mainly by activated macrophages and neutrophils. It is involved in a broad range of cellular activities, including proliferation, survival, differentiation, and apoptosis. The TNF ligand family functions via interactions with its cognate membrane receptors, including the TNF receptor (TNF-R) family [47,48].

Two specific cell surface receptors, tumor necrosis factor receptor superfamily members 1A and 1B (TNFRSF1A and TNFRSF1B), can initiate distinct or overlapping signal transduction pathways, resulting in a large spectrum of cellular responses, including normal functions such as immune responses, hematopoiesis, morphogenesis, and cell death [49]. TNFRSF1A and TNFRSF1B, once stimulated, recruit accessory proteins via interactions with their cytoplasmic domains, including TNFR-associated factors (TRAFs), FAS-associated death domains (FADDs), and TNFR-associated death domains (TRADDs). Different signal transduction pathways activate NF-κB [50].

TNF-α, TNFRSF1, TNFRSF2, TNF-N, CD40L, FASL, and 4-1BBL have all been identified in zebrafish. Zebrafish TNF-α is required for several functions, such as larval fin and spinal cord regeneration [21,51,52], and development of the retina [53], liver [54], and blood vessels [55]. TNF-α promotes oligodendrogenesis after myelin injury in a NF-κB–dependent manner, and zebrafish mutants for myeloid differentiation factor 88 (Myd88), a key adaptor for the activation of NF-κB signaling, result in reduced generation of TNF-α in lesions [44]. TNF-α plays a key inflammatory role in Duchenne muscular dystrophy (DMD). In a well-characterized DMD zebrafish embryo model [56], 1,3-1,6 β-glucans showed a significant effect on the inflammatory state of DMD, as shown by the downregulation of the TNF-α cytokine [57].

The proinflammatory effects of zebrafish TNF-α can be mediated through the activation of endothelial cells [28], and a balance of signaling mediated by TNFRSF1A or TNFRSF1B receptors is required for endothelial cell integrity [55]. Genetic depletion of TNFRSF1B in zebrafish embryos results in the induction of apoptosis, which is rescued by simultaneous depletion of TNFRSF1A or activation of NF-κB [55].

The dual effects of TNF-α have also been described in the context of tuberculosis [12,58,59]. Excess TNF-α, via induction of reverse electron transport (RET) in mitochondrial complex I, drives the production of mitochondrial reactive oxygen species (mROS) in *Mycobacterium marinum*-infected macrophages, which rapidly triggers necrosis [12,59]. These findings in zebrafish larvae infected with *Mycobacterium marinum* have revealed the mechanisms underlying host resistance versus susceptibility to TNF-α.

A study on the toxicity of heavy metals in adult zebrafish revealed that the expression of the *tnfa* and *il1b* genes increased progressively following exposure to heavy metals [60]. It was also shown that exposure to acute combined severe stress in adult zebrafish results in persistent behavioral changes that become apparent one week later, along with elevated cortisol levels and increased expression of gene markers in the brain related to activated neuroglia (M1/M2 and A1/A2 imbalance), proinflammatory cytokines including *tnfa*, apoptosis, and key epigenetic enzymes [61].

### 2.3. IL-1β

IL-1β is a potent proinflammatory cytokine produced as an inactive precursor that is processed by caspase-1 and inflammasomes to produce mature IL-1β. Inflammasomes are multimeric complexes that generally comprise one member of a family of Nod-like pattern recognition receptors (NLR), the adapter protein Pycard, and Caspase-1, which autoactivates and cleaves IL1β in response to a variety of microbial and host-derived stimuli [62,63,64]. In particular, inflammasome activation takes place in two sequential steps: a “priming” signal, usually mediated by Toll-like receptor (TLR) induction, resulting in transcriptional activation of NFκB and synthesis of IL1β and NLRs, and a second “activation” signal, which is responsible for the assembly and activation of caspase-1 to cleave IL1β into its active form [62,63]. Once secreted, mature IL-1β binds to the Interleukin-1 receptor, type 1, or IL1R1, whose conformational change leads to the binding of the co-receptor IL-1 receptor accessory protein, or IL1RAP. The resulting trimeric complex induces a strong proinflammatory signal [65].

Inflammasome-forming proteins are present in zebrafish [66,67,68], and zebrafish Caspase a (Caspa) has the ability to cleave IL1β [69]. Moreover, utilizing a genetic zebrafish model of Il-1β-induced inflammation [70] in combination with in silico analyses, it was recently identified that functional zebrafish Il1r1 has predicted protein structures highly similar to human IL1R1 [71].

The ease of genetic manipulation in zebrafish has given rise to several transgenic models that help to dissect the role of IL-1β. Examples include heat-shock-inducible mature IL-1β [72], cell-specific (pancreatic β cells) expression of mature Il-1β [73], and recently, a doxycycline-inducible model driving expression of *il1b* together with other two inflammatory cytokines: *tnfa* and *ifng1* [74]. These systems highlight IL-1β as a key factor for the recruitment of neutrophils, but not macrophages, to the injury-induced inflammatory site [72] as an inducer of islet inflammation [75]. IL-1β and TNF-α are involved in spinal cord regeneration, tissue injury, and regeneration [76,77].

In adult zebrafish, it has been demonstrated that IL-1β is involved not only in infection but also in the sterile inflammatory response [78]. The zebrafish liver exhibits an accumulation of lipids and displays abnormalities when exposed to ethanol in water, which is accompanied by an increase in *il1b* expression, suggesting that inflammatory signaling is a key factor in hepatic steatosis [79].

### 2.4. Notch

Some of the same inflammatory stimuli, such as IL-1β and TNF-α, which are responsible for the activation of inflammatory signaling, can also induce Notch signaling. The latter does not have a specific pro- or anti-inflammatory effect but can lead to the activation of inflammatory signaling. Indeed, the crosstalk between Notch and the inflammatory compartment has been reported to be key in both development and aging [80,81,82].

Notch is a transmembrane receptor, and its signaling is established through cell-cell contact. One of the four transmembrane Notch receptors (Notch1, Notch2, Notch3, and Notch4 in mice; Notch1a, Notch1b, Notch2, and Notch3 in zebrafish) in a signal-receiving cell binds to Jagged and Delta ligands in a signal-emitting cell and undergoes two cleavage events. First, the Notch receptor is cleaved by members of the ADAM TACE metalloproteases at the S2 site, followed by γ-secretase at the S3 site to release a Notch intracellular domain (NICD) that translocates to the nucleus to modulate the transcription of Notch target genes [83].

Zebrafish Notch1a and Notch1b receptors are evolutionary paralogues of mammalian Notch1 [84], and, as in mammals, zebrafish Notch signaling is involved in both organ formation and morphogenesis and in cell-cell communication [85,86]. Thus, the signals exchanged between neighboring cells through the Notch pathway can modulate differentiation, proliferation, and apoptosis, thereby influencing cell fate and tissue homeostasis [87].

Zebrafish Notch signaling was described to be involved in larval zebrafish heart regeneration [81] and in the establishment of arterial system development and arterial-venous identity [88]. The Notch pathway controls cell fate decisions during embryonic development [89,90]; activation of Notch is required for initiating endothelial cell and Hemogenic Endothelium formation in the dorsal aorta [91] and for regulating goblet cell numbers in the developing zebrafish intestine [92].

In adult aging and regeneration, the endocardium stimulates myocardial regeneration by delivering negative and positive proliferative signals through the Serpine1 and Notch pathways, respectively [93]. Some studies have used adult zebrafish as animal models to discern the role of Notch in eye regeneration. In a zebrafish model of inherited retinal dystrophy, inhibition of the Notch pathway stimulated photoreceptor regeneration in models of progressive degeneration, and immunosuppression prevented photoreceptor loss [82]. Another study reported that Notch signaling is a key pathway for Muller-Glia reprogramming in zebrafish [94]. These findings provide valuable insights into the regenerative processes that rely on Müller glia and the impact of the inflammation-Notch pathway axis on photoreceptor degeneration.

In general, there are inflammatory signals that, in the context of development, are beneficial, but in adults, they are mainly detrimental (Figure 1). Table 1 summarizes the studies reported above that were performed on larvae and adult zebrafish classified by the inflammatory pathway affected.

## 3. Inflammation as a Developmental Mechanism for HSPC Emergence

Hematopoiesis is the process that gives rise to blood cells of different lineages throughout normal life. The hematopoietic system is organized as a hierarchy of cell types that gradually lose multiple alternate potentials while committing to lineage fate. Hematopoietic stem and progenitor cells (HSPCs) can commit to either myeloid or lymphoid lineages, giving rise to innate and adaptive branches of the immune system, respectively [95,96,97,98].

The ontogeny of the hematopoietic system relies on a specific sequence of events and spatial and temporal cellular interactions throughout development, which control stem cell properties and balance self-renewal, quiescence, and lineage commitment [99,100]. As such, hematopoiesis is a complex and accurate developmental process that provides a good platform to identify the genetic requirements of specific critical stages and characterize the molecular mechanisms and cellular mediators of developmental inflammation [101,102,103].

Zebrafish have provided a superb model system to dissect a variety of inflammatory cues specifying HSPCs during the early stages of embryonic development [89,103,104,105], as summarized in Figure 2.

### 3.1. Zebrafish Hematopoiesis

Zebrafish are at the forefront of vertebrate model systems for studying vertebrate developmental hematopoiesis. The high homology of cellular components, core regulators of hematopoiesis, and conserved expression and function of genes for both myeloid/lymphoid fate determination and terminal differentiation between humans and zebrafish make them a powerful model for studying the developmental process of hematopoiesis [101,106,107,108,109,110]. Moreover, defects in zebrafish hematopoiesis reliably phenocopy human blood disorders, including leukemia, making them an attractive model for defining critical regulators of these blood disorders [111,112]. Given these advantages, studying zebrafish hematopoiesis has led to fundamental knowledge and crucial new discoveries regarding different aspects of normal and pathological blood development.

HSPCs arise from the posterior lateral mesoderm (PLM) [113], emerging from specialized endothelial cells (ECs) that compose the hemogenic endothelium (HE) found within the ventral wall of the dorsal aorta (DA) (equivalent to the AGM in mammals). The process of HSPC conversion from ECs is termed the endothelial-to-hematopoietic transition (EHT) and involves the budding of HSPCs/HPCs from the aortic endothelium [114,115]. Nascent HSPCs then enter circulation to migrate from the DA to the caudal hematopoietic tissue [116,117], where they expand and differentiate into mature blood cells. The majority of HSPCs then re-enter circulation and seed their final destination in the thymus or kidney marrow (equivalent to the bone marrow in mammals).

### 3.2. Sources of Developmental Inflammation in HSPC Emergence and Specification

The consensus dictates that the developmental inflammation responsible for HSPC emergence and specification is sterile. In the absence of any pathogen or pathological condition, sources of inflammatory signals during development are, at least in part, hematopoietic cells derived from the earlier waves of hematopoiesis, primarily myeloid cells such as primitive macrophages or neutrophils [118,119]. These primitive myeloid cells can induce a hemogenic cell fate (runx1+; gata2+) in the vascular endothelium (kdrl+; fli1+) and are in close proximity to emerging HSPCs in the HE of zebrafish, mice, and humans [120,121,122]. Other sources of inflammatory signals in sterile setups have recently been summarized [101]. Among these, metabolic alterations due to glucose exposure are responsible for the activation of the NLRP3 inflammasome complex, which in turn elicits sterile inflammation and HSPC emergence [102]. Elevated glucose levels accelerate the induction of HSPCs from the hemogenic endothelium via a Hif1α-regulated signaling axis [123,124]. HSPC generation is also induced by the release of adenosine from ATP, and during shear stress generated by blood flow [125], zebrafish mutants of the adenosine receptor have decreased Cxcl8/Cxcr1 signaling, resulting in reduced EHT and diminished HSPC output [102].

Repetitive element transcripts and R-loops also participate in the induction of developmental inflammation [101]. Transposable elements expressed in the zebrafish hemogenic endothelium can induce NF-κB and IFN-mediated inflammatory pathways by activating retinoic acid-inducible gene 1-like receptors (RLRs). Indeed, Rig-1 and Mda5 zebrafish RLRs mutants exhibit impaired HSPC production [126]. Finally, loss of the DEAD-box helicase Ddx41 triggers an R-loop-mediated sterile inflammatory cascade, characterized by the accumulation of R loops or RNA:DNA hybrids and the induction of the cGAS-STING inflammatory pathway, leading to increased numbers of hemogenic endothelium and HSPCs [127].

### 3.3. Cytokines Role in HSPC Emergence

Primitive macrophages and neutrophils are the main source of proinflammatory cytokines, such as IL-1β and TNF-α, as previously described. These cytokines act mainly downstream of Notch and NF-κB. Other cytokines involved in HSPC specification and emergence include IFN-γ [128], IL6 [129], TGFβ [130], and Gcsfa/Gcsfb [131].

It has been previously established that glucose metabolism expands HSPC formation in zebrafish embryos through mitochondria-derived ROS-mediated stimulation of hypoxia-inducible factor 1α (hif1α) [123]. Recently, it was reported that metabolic alterations promote inflammasome-induced IL1β signaling to enhance HSPC production. Macrophages are the main source of this cytokine, and they promote the production of Il1rl1+ HSPCs via inflammasome action, with loss of inflammasome function inhibiting HSPC production in zebrafish embryos [102].

### 3.4. NF-κB Signaling in HSPC Emergence

Canonical NF-κB activation is required for HSPC specification during embryogenesis [118]; indeed, the specific impairment of nuclear NF-κB translocation in the hemogenic endothelium impairs NF-κB activation and further HSPC emergence [118,119]. Mechanistically, NF-κB regulates hemogenic endothelium-derived HSPC development signaling by promoting Notch activity [119], and primitive neutrophils produce TNF-α, which in turn stimulates NFκB-dependent expression of the Notch ligand jagged1, *jag1*, in *tnfr2*-expressing endothelial cells of DA. Jag1 binds to its receptor Notch1a on neighboring hemogenic endothelial cells lining the DA floor, which in turn activates runx1, enforcing hematopoietic cell fate [118]. Conversely, in the absence of jagged1, zebrafish fail to produce HSPCs [132] and display an established arterial fate, but have compromised HE and HSPC formation [130].

The involvement of NF-kB in EC priming towards the fate of HE has been described recently. Cheng and colleagues found that the non-inflammasome-forming NLR Nod1 programs the endothelium to become hemogenic, initiating the HE commitment, which is a prerequisite for HSPC specification. They also observed a remarkable downregulation of NF-kB activity in the DA at 22 h post-fertilization in embryos deficient for either Nod1, Ripk2, or Rac1. Further characterization of this pathway led them to identify the axis Rac1-Nod1-Ripk2 to signal through NF-kB in order to initiate HE commitment [133].

### 3.5. Notch Signaling in HSPC Emergence

One essential requirement for HSPC emergence in both mammals and zebrafish is signaling through the Notch pathway [83,89,134,135,136]. Here, we summarize some studies concerning the role of Notch in HSPC development. However, given the strong connection with both stimuli and targets of the inflammation compartment, we believe further studies should address the crosstalk and integration of Notch with other signaling pathways in order to provide a better understanding of the developmental signalosome in the hematopoietic niche.

In zebrafish, both Notch1a and Notch1b receptors are expressed in the DA during the window of HSPC emergence [88], and the induction of Runt-related transcription factor 1, or Runx1, one of the earliest markers of HSPCs [114,137], relies on Notch1a/b receptor-mediated signaling in the HE [83].

Runx1 is highly expressed in endothelial cells and is positive for the Notch reporter line Tp1:GFP, which expresses GFP under the control of tandem Notch responsive elements [138]. Interestingly, HE identity may be established by Notch signaling much earlier than the onset of *runx1* expression [83,89,139].

The requirement for Notch-Runx1 signaling is also a key feature distinguishing HSPCs from erythro-myeloid hematopoietic progenitors (EMPs) [140,141]. HSPCs are unable to form in the absence of Notch-Runx1 signaling; in contrast, EMPs do not express the Notch receptor, and their generation is not affected under these conditions [140,141]. Moreover, induction of the Notch intracellular domain (NICD) in wild-type zebrafish embryos leads to upregulation of HSPCs without inducing ectopic arterial gene expression [110,140,142]. When the interaction of Notch ligands DeltaC and DeltaD with Jam1a and Jam2a on early vascular progenitors is missing, arterial specification is unaffected, but HE is lost [143], further proving evidence of the Notch-specific role in HSPC generation from the HE.

### 3.6. Demand-Driven Hematopoiesis Influences HSPC Lineage Commitment

The role of developmental inflammation in HSPC lineage commitment and blood cell maturation and differentiation is still poorly characterized. Whether baseline inflammatory signaling might favor one lineage over another is not explored. However, once in demand, that is, in response to pathogens or any external cue, inflammatory signals might change the programmed fate of HSPCs by subverting the proliferation and/or differentiation of HSPCs or progenitor cells. This is the case of emergency myelopoiesis or lymphopoiesis [144,145].

## 4. Zebrafish as a Model to Study Inflammaging

Life expectancy has increased over the past few decades. According to the World Health Organization, more than 10% of the world’s population is over 60 years old, and by 2050, it will nearly double. Zebrafish is a promising model in the field to study different age-related diseases due to its genetic and physiological similarities to humans [5,146,147].

It has been described that the aging process is usually accompanied by a decline in immune function, causing chronic inflammation in older organisms. Age-related inflammation is called “inflammaging” [148,149,150]. Recent studies using zebrafish have shown that microglial density is higher in aged zebrafish than in young adults [151]. In addition, 18-month-old animals reported fewer microglia and macrophages in the optic nerve at four days post-myelin damage compared to young adults [152].

### 4.1. Telomeres

Telomeres are hexanucleotide tandem repeats of DNA and associated proteins that form dynamic structures at the ends of chromosomes, which can maintain genomic stability and integrity. Telomeres are maintained by the telomerase ribonucleoprotein, and loss of telomerase function leads to telomere shortening and chromosomal instability, ultimately leading to aging and cancer [153,154]. It was shown that zebrafish can replicate human telomere and telomerase biology [7,155]. Zebrafish lacking telomerase are valuable models for investigating telomere-driven aging. Furthermore, it is an outstanding vertebrate model for the development of new therapies that can temporarily restore telomerase expression in individuals with diseases with partial telomerase deficiency, such as dyskeratosis congenita and aplastic anemia [156,157].

Age-associated telomere shortening affects the adaptive immune system, leading to immunosenescence [158]. Despite the decline in immune function, research has shown that older organisms often experience a persistent proinflammatory state, known as inflammaging. This condition is characterized by elevated levels of proinflammatory markers in cells and tissues as well as chronic activation of the innate immune system, which can occur even in the absence of infection or other risk stimuli [149].

Other studies using telomerase retrotranscriptase (*tert*)-mutant zebrafish have shown that telomere shortening increases senescence and systemic inflammation, facilitating melanoma dissemination [159]. As a result, similar to the aging process in humans, telomere shortening creates a persistent inflammatory environment, which, in turn, contributes to a higher incidence of cancer. There have been described extracurricular roles of the internal RNA template component of the telomerase complex (*TR* or *terc*), regulating the levels of the cytoquines Gcsfa/Gcsfb and the key transcription factors *spi1* and *gata1*, which are responsible for controlling the production of myeloid and erythroid cells, respectively [160]. Additionally, telomerase RNA-based aptamers can restore defective myelopoiesis in zebrafish models of congenital neutropenic syndromes [161]. Aplastic anemia and dyskeratosis congenita patients present mutations in *TR* and *TERT* [162,163]. These models of *tert* and *terc* deficiencies in zebrafish may be useful for identifying potential therapeutic approaches for treating these diseases.

### 4.2. TERRA

In recent years, long fragments of non-coding RNA expressed from telomeres, called TERRA, have been identified [164,165]. These form part of the nucleoprotein complex, promoting chromosomal stability and the replication of telomeric repeats [166,167]. It has recently been reported that telomere dysfunction can lead to the activation of inflammatory signaling via secretion into the extracellular environment of TERRA within exosomes, known as cfTERRA [168].

Activation of the telomerase maintenance mechanism is an essential step in cancer progression to escape senescence and apoptosis and is routinely performed by telomerase retrotranscriptase. Alternative telomere lengthening (ALT) associated with increased TERRA expression has been observed in certain malignant tumors. The expression of *tert* has been shown to reverse ALT characteristics and normalize TERRA expression levels in zebrafish [169]. These findings indicate that TERRA and telomerase exhibit opposing functions in telomere maintenance through several mechanisms, as the reintroduction of tert in brain tumors promotes the formation of heterochromatin and decreases the levels of TERRA, leading to enhanced survival in fish and opening up possibilities for future treatments for ALT brain tumors.

### 4.3. Senescence

Another zebrafish aging model is *the rag1* mutant. Recombination-activating gene 1 (RAG1) plays an essential role in adaptive immunity, orchestrating DNA rearrangements that generate an enormous variety of immunoglobulins [170]. Zebrafish have one functional *rag1* gene, and its loss of function leads to a deficiency in the functional adaptive immune system [171]. Unlike mammals, *rag1* mutant zebrafish do not show increased susceptibility to infection and respond more quickly to viral infections [172]. However, *rag^−^*^/*−*^ zebrafish exhibit early signs of aging and a reduced lifespan [173]. It has been shown that *rag1*-deficient zebrafish have increased immune-related gene expression and oxidative stress due to a decline in antioxidant activity and increased oxidative cell damage. Moreover, the number of senescent cells increased, and telomere length was shorter than that in wild-type zebrafish. The use of antioxidant and senolytic drugs, which induce apoptosis in senescent cells, reduced interleukin 1b and senescence gene expression in rag^−/−^ zebrafish.

Recently, it was reported that senolytic drugs have anti-inflammatory effects in zebrafish models of chronic skin inflammation and in a high-cholesterol diet [174,175,176,177,178,179]. Taken together, these characteristics suggest that rag mutant zebrafish offer a valuable platform for investigating and seeking treatments for senescence in vivo, as well as for the aging process induced by chronic inflammation (Figure 3).

Despite all these advances, more research is needed to understand the complex interactions among metabolism, the immune system, and aging, and zebrafish is emerging as a key model for studies. In a broader context, these studies underscore the significance of using zebrafish to assess drug efficacy across various disease models. A summary of the different disease models developed in the zebrafish is presented in Table 2.

## 5. Conclusions

This review provides a comprehensive overview of the significance and utility of the zebrafish model system for investigating inflammation throughout the developmental and aging processes. Here, we review the main inflammatory pathways examined in zebrafish and summarize the available knowledge on their function in both development and aging. Specifically, we present examples illustrating how different cell entities rely on inflammatory signals to enable their specification and function during development. For example, modeling developmental inflammation in larval zebrafish has provided unique insights into the role of inflammation in the regulation of steady-state hematopoiesis. In contrast, in aging, inflammation is often a chronic condition fostering a pathological onset, called inflammaging, and could be linked to age-related diseases. These studies provide evidence that the same inflammatory signaling pathways often have instructive/beneficial roles during embryogenesis, whereas they are linked to malignancies in adults. Thus, the dichotomy of whether the same stimuli and signaling pathways are instructive or malignant depends on the developmental stage. As reviewed here, zebrafish offers a powerful toolkit to answer these questions and is a unique, reliable system for drug discovery.

## Figures and Tables

**Figure 1 ijms-25-02145-f001:**
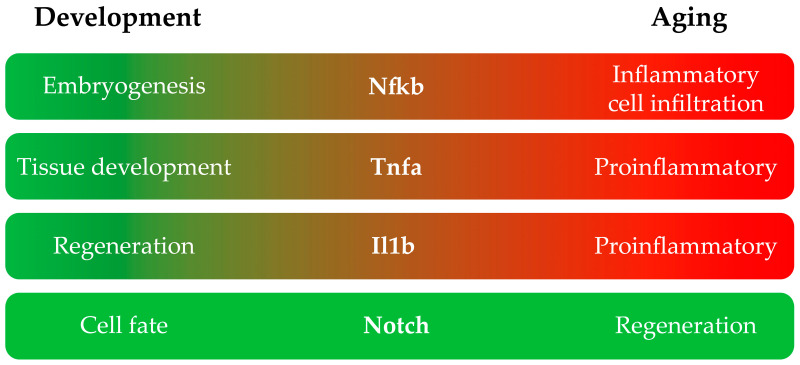
Main roles of each inflammation pathway in the development and adult stages of zebrafish.

**Figure 2 ijms-25-02145-f002:**
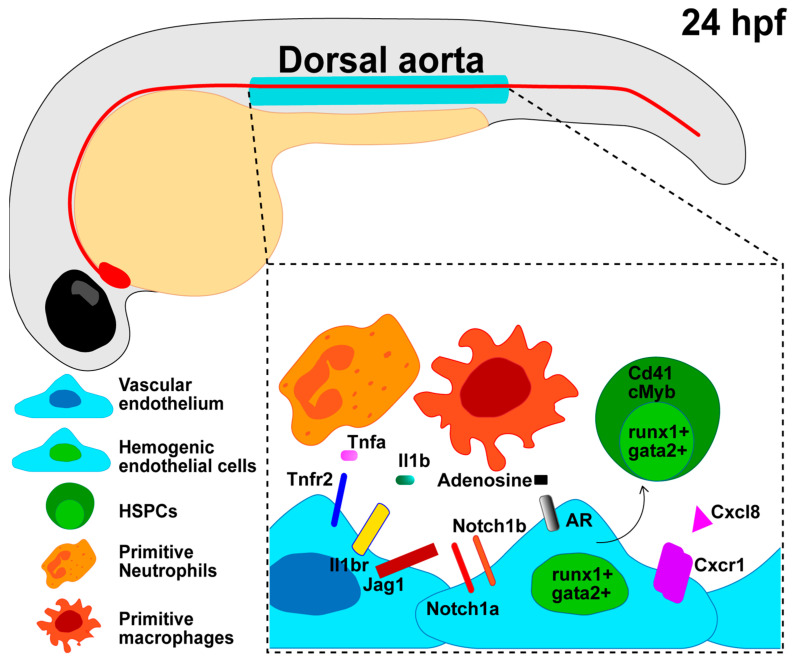
Developmental inflammation regulates the endothelial cell transition into HSPCs. Zebrafish HSPCs emerge around 24 h post-fertilization (hpf) from a specialized endothelial cell termed the hemogenic endothelium (HE) in the ventral wall of the dorsal aorta. Notch, Nfkb, and other inflammatory signaling pathways can induce a hemogenic cell fate (runx1+; gata2+) in the vascular endothelium (kdrl+; fli1+) and further HSPC commitment (runx1+; gata2+; Cd41, cMyb). Sources of developmental inflammation include cytokines released by primitive macrophages or neutrophils that interact with the vascular endothelium and HSPCs, as well as glucose exposure, the release of adenosine from ATP, repetitive transcript elements, and R-loops.

**Figure 3 ijms-25-02145-f003:**
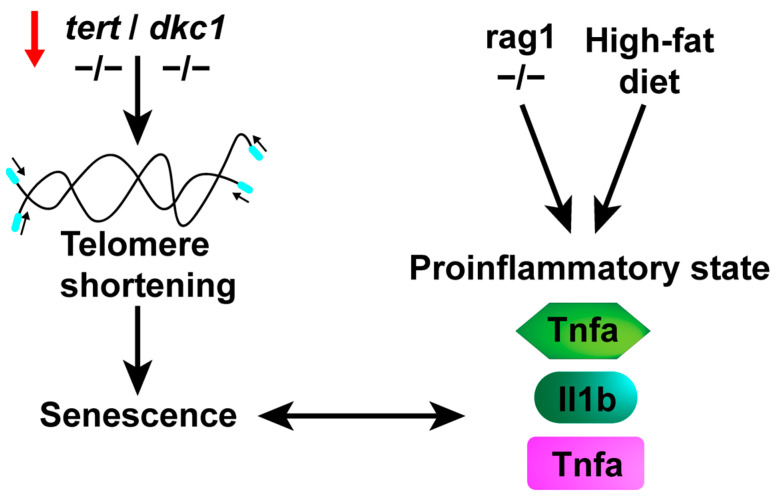
A proinflammatory state is triggered by stimuli such as a high-cholesterol diet and/or the aging process. Telomere shortening, associated with aging or various diseases affecting telomerase components, leads to cellular senescence. This progression results in an elevation of proinflammatory markers, giving rise to a proinflammatory state. Additionally, congenital deficiencies or a high-cholesterol diet can also induce this state, ultimately culminating in senescence.

**Table 1 ijms-25-02145-t001:** Inflammation pathways involved in the development and adult stages of zebrafish.

Inflammatory Signaling Pathway	Development	Adult
NF-κB	Antiviral responses [43]	Pancreatic beta-cells proliferation [6]
	Oligodendrogenesis after myelin injury [44]	*sirt1* knockout leads to oxidative injury, chronic inflammation, and a reduced life span [45]
	Mesoderm development and embryonic dorsalization [42]	
TNF-α	Fin regeneration [21]	Heavy metal toxicity [60]
	Retinal neurogenesis and optic myelination [53]	
	Liver development [54]	Delayed responses to acute stress [61]
	Blood vessel development [55]	
	Oligodendrogenesis after myelin injury [44]	
IL-1β	Fin regeneration [77]	Inflammatory response in steatosis [79]
Inflammatory compartment via Notch	Cardiomyocyte proliferation and heart regeneration [81,93]	Regeneration of Inherited Retinal Dystrophy [82]
	Arterial system development [88]	Endocardium regeneration [93]
	Endothelial cell and Hemogenic Endothelium formation [91]	Müller-Glia-mediated retinal regeneration [94]
	Intestinal goblet cell homeostasis [92]	

**Table 2 ijms-25-02145-t002:** Zebrafish models of inflammatory, developmental, and aging diseases.

Model	Generated by	References
Duchenne Muscular Dystrophy	*sapje* mutation	[56,57]
Myelinogenesis	*cd59^uva48^* mutation	[46]
Psoriasis and atopic dermatitis	*spint1a* mutation	[180,181,182,183,184]
Noonan Syndrome-Myelomonocytic leukemia	Shp2*^D61G^* knock-in	[185]
Chronic inflammation and oxidative injury	*sirt1* knock-out	[45]
IBD and MAFLD/MASH	High-cholesterol diet	[174,179,186]
EAE multiple sclerosis	Regimen of MOG	[187]
Skeletal muscle atrophy	Ethanol exposure	[15]
Model of Inherited Retinal Dystrophy	Eye lesion	[82]
Heavy metal induced toxicity	Immersion in heavy metal	[60]
PTSD and Stress related disorder	Stress stimuli	[61]
Alcohol-induced steatosis	0.05 v/v ethanol	[79]
Systemic inflammation	Il1b induced secretion	[70]
Dyskeratosis congenita	*tert/terc/dkc1* knock-out	[155,159,160,188,189]
Severe combined immunodeficiency	*rag1* mutation	[173]

IBD: Intestinal bowel disease MAFLD: Metabolic-associated fatty liver disease; MASH: Metabolic-associated steatohepatitis PTSD: Post-traumatic stress disorder.

## Data Availability

Not applicable.

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
