# Peer review of "Inflammation in Development and Aging: Insights from the Zebrafish Model"

_ijms, 2024, doi:10.3390/ijms25042145_

Round 1
Reviewer 1 Report
Comments and Suggestions for Authors
The manuscript effectively leverages the unique advantages of the zebrafish model in exploring the multifaceted roles of inflammatory pathways in development and aging, offering robust support for modeling human diseases. The manuscript distinctly emphasizes the significance of zebrafish as a model for human diseases, particularly due to its significant genetic similarity to humans, rapid development, and ease of genetic manipulation. It provides valuable insights into the understanding of inflammation in zebrafish development and aging.
Overall, I recommend accepting the manuscript for publication.
Author Response
Dear Reviewer,
Thank you for your positive feedback and for recommending our manuscript for publication. We are pleased to hear that you found our work on the multifaceted roles of inflammatory pathways in zebrafish development and aging valuable. We appreciate your recognition of the unique advantages of the zebrafish model and its significance in modeling human diseases.
We look forward to seeing our work published and hope that it will provide a novel perspective of inflammation in development and aging.
Best regards.
Reviewer 2 Report
Comments and Suggestions for Authors
The authors that summarized the recent results in zebrafish model system that are focused in inflammatory response during early developmental stage, tissue development and adult stage, which correlated to aging process and some diseases.
It is interesting, some comment as following:
1. In title should be try to modify as Inflammation in an early embryonic, tissue developmental and adult aging process in zebrafish model.
2. The authors should provide a summarized figure to mention (1) inflammation in early development stage, (2) inflammation in tissue development stage, and (3) inflammation in adult stage that correlated to aging process.
3. In figure 1 should be checked the spelling such as Tnfr2 and Tnfa as a Tnfγ2 and Tnfα.
4. In Table 2, should be checked the abbreviation such as IBD: Intestinal bowel disease; MAFCD: Metabolic-associated fatty liver disease; …
Comments on the Quality of English LanguageMinor editing of English language required
Author Response
Dear reviewer,
Thank you for your valuable feedback. We appreciate your time and effort in reviewing our manuscript. Please find our responses to your comments below:
- We greatly appreciate your insightful comment, which has provided us with an opportunity to refine the scope and content of our review. In response, we have made minor adjustments throughout the manuscript to enhance clarity in our definitions of development and aging. Additionally, we have improved the introduction by explicitly outlining the elements that will be discussed in the text. After thorough consideration, we have chosen to retain the original title, as we believe it accurately represents the scope and content of our study.
- We have created a new figure that illustrates the inflammation induction by aging and diet.
- We have changed the spelling of the terms to the correct zebrafish name.
- For clarification, all abbreviations used in the table are explained in the footnote of the table.
We hope these revisions will strengthen our manuscript and make it more suitable for publication. We look forward to hearing from you regarding the next steps.
Best regards.
Reviewer 3 Report
Comments and Suggestions for Authors
Zebrafish are a current and important animal model in biomedical research. Zebrafish provide several advantages to studying inflammatory processes and several pathologies related to human health. The present review presents an interesting perspective on inflammatory research in zebrafish. A general recommendation would be to improve the Introduction or the first section by clearly mentioning which elements are going to be discussed in the text and why the authors will start with NF-κB, TNF-α, IL, etc. This could be improved by giving a brief review of the inflammatory responses in zebrafish and stating if it is similar to humans.
The Introduction and the keywords mention “regeneration”, but little is mentioned about this in the text. I recommend adding more studies regarding regeneration, describing the findings.
Line 15. This might be my personal preference; however, I recommend changing the sentence “Zebrafish are powerful tools for modeling human disease…” to “Zebrafish are an essential animal model to study human disease…”. In this way, it does not sound like zebrafish are research objects. This recommendation also applies to line 31.
Keywords. Consider adding the scientific name zebrafish instead of “zebrafish”.
1. Zebrafish as inflammation model. When mentioning that zebrafish has contributed to several biomedical fields, it would be interesting to briefly mention what has been studied/found by using zebrafish as an animal model. For example, it is mentioned that COVID-2019 was also studied in zebrafish. Here, the authors could add one or two examples of how zebrafish contributed to the study of COVID-19 pathophysiology/pharmacological protocols/etc. Another example is fin injury models. The main findings regarding regeneration and tissue repair in zebrafish could be added as well.
Lines 61-63. I recommend replacing these lines with a brief introduction (a couple of paragraphs) about inflammation, how inflammation starts, and an overview of the main inflammatory substances that have been studied in zebrafish. Also, the similarities between the inflammatory response between humans and zebrafish could be added so the reader can understand why the authors mention NF-κB, TNF-α, IL, etc., in the next subsections.
Table 1. The Table could be improved by adding the objective of each study and the main findings of each.
Lines 235-245. This paragraph needs references.
Lines 454. Before the conclusions, I suggest adding a section where the authors discuss the current field of research where zebrafish can be used to study aging and inflammaging. What else needs to be studied or has been recently reported? Mentioning this is important because using zebrafish as animal models has been increasing in the last few years.
Author Response
Dear Reviewer,
Thank you for your insightful comments and suggestions. We appreciate the time and effort you have put into reviewing our manuscript. Here are our responses to your comments:
We agree with your suggestion to improve the introduction by clearly stating the elements that will be discussed in the text. We revised the introduction to include a brief review of the inflammatory responses in zebrafish and their similarity to humans.
We acknowledge your point about the mention of “regeneration” in the introduction and keywords. The goal of our review is to emphasize the significance of zebrafish as a model for studying inflammation in both development and aging, and to highlight the multifaceted roles of the same inflammatory pathways in these two different “life stages”. We decided to address minor changes over the entire manuscript to be clearer in our definition of development and aging and on the goal of our review. Following your comments we provided more specific examples concerning inflammation signaling in regeneration, however, we did remove these as keywords because not in the focus of our review.
We appreciate your suggestion to change the phrasing in lines 15 and 31 to emphasize that zebrafish are not merely research objects but an essential animal model. We made these changes accordingly.
We added the scientific name of zebrafish to the keywords as per your recommendation.
We expanded the section on zebrafish as an inflammation model to include examples of how zebrafish contributed to the study of COVID-19, and findings regarding the use of the tailfin injury technique for addressing regeneration and tissue repair in zebrafish.
We completed lines 61-63 with a brief introduction about inflammation, how it starts, and an overview of the immune development. We also highlighted the similarities between the inflammatory response in humans and zebrafish.
In Table 1 we provide a concise summary of all the models discussed in the text. To maintain simplicity and readability, we highlight only the main topic of each study in the table.
We added references to the paragraph in lines 235-245 as recommended.
We added a brief reflection before the conclusions as recommended.
Following your suggestions, we hope these revisions improve the quality of our manuscript. We look forward to your further comments and suggestions.
Best regards.